# Real-Time Phase-Contrast MRI to Monitor Cervical Blood and Cerebrospinal Fluid Flow Beat-by-Beat Variability

**DOI:** 10.3390/bios12060417

**Published:** 2022-06-15

**Authors:** Giuseppe Baselli, Federica Fasani, Laura Pelizzari, Marta Cazzoli, Francesca Baglio, Maria Marcella Laganà

**Affiliations:** 1Department of Electronics, Information, and Bioengineering, Politecnico di Milano, 20133 Milan, Italy; giuseppe.baselli@polimi.it (G.B.); federica.fasani@mail.polimi.it (F.F.); 2IRCCS Fondazione Don Carlo Gnocchi ONLUS, 20148 Milan, Italy; lapelizzari@dongnocchi.it (L.P.); mcazzoli@dongnocchi.it (M.C.); fbaglio@dongnocchi.it (F.B.)

**Keywords:** arterovenous flow, cerebrospinal fluid flow, MRI, beat-by-beat variability, rehabilitation, neurodegeneration

## Abstract

Beat-by-beat variability (BBV) rhythms are observed in both cardiovascular (CV) and intracranial (IC) compartments, yet interactions between the two are not fully understood. Real-Time Phase-Contrast (RT-PC) MRI sequence was acquired for 30 healthy volunteers at 1st cervical level on a 3T scanner. The arterial (AF), venous (VF), and cerebrospinal fluid (CSF) flow (CSFF) were computed as velocity integrals over the internal carotid artery, internal jugular vein, and CSF. AF, VF, and CSFF signals were segmented in inspiration and expiration beats, to assess the respiration influence. Systolic and diastolic BBV, and heart period series underwent autoregressive power spectral density analysis, to evaluate the low-frequency (LF, Mayer waves) and high frequency (HF, respiratory waves) components. The diastolic VF had the largest BBV. LF power was high in the diastolic AF series, poor in all CSFF series. The pulse wave analyses revealed higher mean amplitude during inspiration. Findings suggests a possible role of LF modulation of IC resistances and propagation of HF waves from VF to AF and CCSF. PC-RT-MRI could provide new insight into the interaction between CV and IC regulation and pave the way for a detailed analysis of the cerebrovascular effects of varied respiration patterns due to exercise and rehabilitation.

## 1. Introduction

Cardiovascular (CV) beat-by-beat variability (BBV) [1] and cerebral autoregulation [2] have been studied for centuries and decades, respectively. Yet, the interaction between the systemic CV and the intracranial (IC) compartments is still an open question. Short term rhythms found in CV BBV, as Mayer and respiratory waves [3,4,5,6] have been described also in IC BBV [7], suggesting a close link between systemic CV and IC compartments.

Real-time, phase-contrast magnetic resonance imaging (RT-PC-MRI) [8] is a novel advanced magnetic resonance imaging (MRI) technique that allows flow sampling at rates higher than 10 Hz, without gating, allowing BBV monitoring. This pilot study, conducted in a group of young healthy volunteers, addressed arterial flow (AF), venous flow (VF), and cerebrospinal fluid (CSF) flow (CSFF), measured with RT-PC-MRI and their BBVs at the hemodynamic interface between the IC and the systemic CV compartments at the 1st cervical level.

Doppler echography may be an alternative and more accessible approach to non-invasively address the BBV investigation without the constraint of supine position imposed by high-field MRI scanners [9]. However, it suffers from many limitations. First, the intraspinal compartment and CSF cannot be reached with Doppler echography [10]. Second, the ultrasound-derived measures are operator-dependent because the vessels have to be manually targeted by the sonographer. Moreover, it is difficult to keep a steady insonation plane during long recordings. As to flow quantification, some studies refer to the time trace of pulsed-wave Doppler velocity, extracting only points of interest such as the peak velocity [11]. Therefore, compared to Doppler echography, RT-PC-MRI allows for non-invasively assessing BBV in arterial, venous, and liquor flows to and from the IC compartment.

The aims of the current work were twofold. First, proposing a new investigation tool relevant to the interactions of CV BBV and IC BBV. The method provides quantitative insight only at the hemodynamic side of flows. Nonetheless, this approach might be a first step to disentangle the hemodynamic mechanism from the neurological ones in suitable future experiments. Second, quantifying venous drainage and the influence of respiration on it. This is foreseen to impact on rehabilitation protocols for neurodegenerative patients, in which detrimental effects of hindered venous outflow were observed [12,13].

In addition, despite being beyond the scope of this work, it is worth mentioning a potential impact on hemodynamic modelling studies, i.e., the modelling of AF, VF, and CSFF and their interactions in the IC circulation [14,15,16,17,18,19]. In this field, moving from the fitting of single gated wave shapes to several sampled ones is expected to highly increase the statistical power of model fitting to be exploited by parametric identification.

## 2. Materials and Methods

### 2.1. MRI Acquisitions

Thirty healthy volunteers (age range: 19–58 years; 21 males) were acquired on a 3T MRI scanner (MAGNETOM Prisma, Siemens Healthcare, Erlangen, Germany) equipped with a 64-channel head-neck coil. The study was approved by IRCCS Fondazione Don Carlo Gnocchi Ethics Committee, and it was performed in accordance with the principles of the Helsinki Declaration. Written and informed consent was obtained from all the participants.

A prototype RT-PC sequence based on segmented echo-planar imaging (EPI) readout, parallel temporal acceleration factor, and 2-sided shared velocity encoding a reconstruction algorithm, was used to measure blood flow and CSFF at the first cervical level (field of view [FOV] = 153 × 175 mm^2^, matrix size = 96 × 128, interpolated resolution = 0.7 × 0.7 mm^2^, slice thickness = 8.6 mm). 

For the AF and VF measurement, the imaging slice was placed perpendicular to the main neck vessels (i.e., internal carotid arteries and internal jugular veins), with the following parameters: temporal resolution = 58.5 ms (17.094 Hz), velocity encoding [VENC] = 70 cm/s, GeneRalized Autocalibrating Partial Parallel Acquisition [GRAPPA] = 3, repetition time [TR]/echo time [TE] = 14.6/8 ms, flip angle = 15°. For the CSFF quantification, the imaging slice was placed orthogonal to the spinal cord, with temporal resolution = 94 ms (10.638 Hz), VENC = 6 cm/s, GRAPPA = 2, TR/TE = 15.7/9 ms, flip angle = 5°.

Subjects were scanned in resting state. Claustrophobic subjects and subjects experiencing stress due to MRI were excluded from the study, to ensure the acquisition of AF, VF, and CSFF in supine basal condition, at rest. In addition, no caffeine or drugs intake was allowed before the scan.

To assure a steady-state between the two runs, acquisition time was limited to a minimum duration necessary for BBV analysis, which was 60 s (i.e., 60–80 beats). Pulse (finger plethysmography) and respiration (thoracic belt) were also recorded for a quality check relevant to heart rate (HR) and breathing rate (BR), respectively. The latter was also for classifying beats occurring in the inspiration and expiration phases.

### 2.2. MRI Image Processing

RT-PC-MRI frames were analyzed with SPIN software package (SpinTech Inc., Bingham Farms, MI, USA) [20] by a single trained operator. Time frames with the highest flow were selected, and regions of interest (ROIs) corresponding to the internal carotid arteries (ICAs), internal jugular veins (IJVs), and the peridural rachis CSF space were drawn using a semiautomated method. Four regions of static tissue (no-flow areas) were manually drawn close to the ROIs, for background phase correction. Then, ROIs were copied to all time frames and manually adjusted, if needed. Phase images were converted into velocity maps, with offset correction based on the average values in the no-flow areas. Finally, AF, VF, and CSFF were computed as velocity integrals over the respective ROI areas. Sampled AF, VF, and CSFF signals had a sampling rate equal to the frame rate: 17.094 Hz for AF and VF, 10.638 Hz for CSFF. Flows were expressed in mL/s, positive upward, negative downward. So, as a convention, AF is positive while VF is negative. 

### 2.3. Flow signal Processing

First of all, the quality of AF, VF, and CSFF signals derived from MRI data was visually checked and, in a few cases, windows of bad signal were discarded. A bad signal had to be mainly ascribed to the combined effect of the subject’s movements and automatic ROI propagation. When copying ROIs in all the frames, in the same position, we assume that the subject is not moving across the whole acquisition; however, if the subject moves in one or more time points of the acquisition, the flow of the structure of interest (ICAs, IJVs, or CSF) is partially or completely outside the copied ROI. This creates a sudden and artefactual flow decrement that can be easily detected looking at the flow rate periodic oscillations (exemplificative signals with artefacts in Figure 1). 

The beat-by-beat analysis of AF and VF was referred to as the diastolic AF flow series *AF_D_*, namely the wave composed of AF local minima, representing the end of diastole (Figure 1, top panel, “*” markers). Although the diastolic VF value should correspond to the maximum value in a cycle (i.e., downward flow minimum), in this study *VF_D_* series were sampled synchronously to the *AF_D_* (Figure 1, mid panel, “*” markers), because VF local maxima were highly modulated and noisy and would have provided a less smooth description of the VF upper envelope. Arterial and venous systolic series *AF_S_* and *VF_S_* were defined as the series of AF local maxima and VF local minima, respectively. The time interval between two neighboring samples of the *AF_D_* was computed to derive the heart period (HP) series, which were assumed to be the HP estimate.

CSFF, which was acquired with a separate sequence, was separately processed. The CSFF diastolic series (*CSFF_D_*) and the systolic one (*CSFF_S_*) were defined as the series of CSFF local maxima (positive, upward) and minima (negative, downward), respectively. End-diastole-to-end-diastole *HPs* were computed also on the CSFF signal, as a double-check, and named *HP_CSF_*.

In addition, mean value series of AF, VF, and CSFF over the signal-specific HP were computed (*AF_M_*, *VF_M_*, and *CSFF_M_*).

### 2.4. Pulse Shape Analysis

AF, VF, and CSFF heart cycle dynamics (alias, pulse waves) were stretched or compressed to the mean HP and normalized to 0–100%. This two-fold normalization was performed to permit both intrasubject and group alignment, respectively. Median and quartile pulse waves were derived for the group. 

The same analysis was repeated segmenting the inspiration and expiration phase cycles, to investigate changes of pulse shapes related to the respiratory phase and to emphasize respiratory modulation with no need of a priori hypotheses (e.g., linearity). AF, VF, and CSFF signals were segmented in inspiration and expiration beats, based on the thoracic belt respiratory signal. This respiratory phase analysis was limited to 27 subjects, since the remaining 3 subjects had bad thoracic belt recordings. 

### 2.5. Beat-by-Beat Variability Spectral Analysis

Spectral analysis of BBV was performed by means of the Autoregressive (AR) Power Spectral Density (PSD). The AR model order was set to 9, as the optimal order defined by means of Akaike’s Information Criterion. 

Each series underwent a linear detrend, resulting in zero-mean BBV series. In case of missing time windows, separate AR identifications were performed on the two or three available tracts of the series. Next, the weighted average of AR parameters was computed considering the respective number of beats as weights.

Then, PSD computation and decomposition (residuals method) was performed. The central frequency (i.e., AR pole phase) and the power (i.e., pole residual) were provided in the very low-frequency range (VLF, 0–0.04 Hz); in the low-frequency range (LF, 0.05–0.15 Hz, alias Mayer waves frequency range); and in the high-frequency range (HF, centered at the breathing frequency given by the thoracic belt registration, with a window of ±0.15 Hz, alias breathing frequency (BF) range). The HF obtained with PSD analysis of MRI-derived signals was compared to the BF derived from the thoracic belt.

The mean values and standard deviation were computed in the group for the following parameters: total power (*P_TOT_*, i.e., series variance, that is PSD area under the curve), VLF power (*P_VLF_*), normalized LF power (*P_LF_*, expressed in normalized units [n.u.]) and normalized HF power (*P_HF_*, expressed in normalized n.u.). Normalized units (n.u.) were defined as (*P_LF_*/*(P_TOT_* − *P_VLF_*))% and (*P_HF_*/(*P_TOT_* − *P_VLF_*))%, for LF and HF, respectively. The subtraction of *P_VLF_* in the definition of n.u [6] is justified by the erratic presence of slow trends in BBV, which in a short-term analysis can be considered close to 0 Hz. Furthermore, VLF effects were further reduced by the linear detrend. Therefore, *P_VLF_* was only used for the computation of n.u. and not further analyzed. It is worthy of note that *P_LF_* + *P_HF_* n.u. may be slightly less than 100% since small (<5% of *P_TOT_*) components due to noise were discarded. 

LF and HF central frequencies (*f_LF_* and *f_HF_*) were expressed in Hz and estimated from the LF and HF pole phase. In the rare case of two or more poles in the same frequency band, the pole barycenter was computed. 

### 2.6. Statistical Analysis

A two-way repeated Measure Analysis of Variance (RM-ANOVA) was used to test power differences due to two factors: (1) the structure where the flow was measured (i.e., arterial, venous, and CSF flow) and (2) the point in the cardiac cycle (systole, diastole). This comparison was performed both for the total power and for the power in LF and HF ranges. A repeated Measure Analysis of Variance (RM-ANOVA) was used to test flow rate differences due to respiration phases (inspiration vs. expiration).

A paired *t*-test was performed to test HP, BR, and *f_HF_* differences between the blood and CSF acquisitions.

All the post hoc comparisons were adjusted for multiple comparisons using the Bonferroni correction. Corrected *p*-values lower than 0.05 were considered significant. 

The observed power (ω) and the effect size estimate Cohen’s dz [21] were computed with G*Power (Düsseldorf, Germany) version 3.1.9.4 [22]. Effect size was interpreted as follows: no effect (dz < 0.2); small (0.2 ≤ dz < 0.5), intermediate (0.5 ≤ dz < 0.8), and large effect (dz ≥ 0.8) based on benchmarks suggested by Cohen [23].

## 3. Results

### 3.1. Extraction of BBV Series

A typical example of AF, VF, and CSFF signals is shown in Figure 1. The asterisks “*” mark the diastolic points. In addition, the systolic points can be easily identified in Figure 1, as the upper envelopes of AF and the lower envelops of VF and CSFF. At a visual inspection, the high level of BBV with large modulation waves is evident, particularly in the VF. CSFF is clearly characterized by a smoother shape. 

An example of a bad signal interval, that was discarded, is also shown (Figure 1, AD and VF panel, grey rectangle). Bad signal intervals occurred in less than 15% of the subjects, and in those cases, less than 10% of the recording (duration = 60 s) had to be discarded.

### 3.2. AR PSD Analysis of BBV Series

An example of AR PSD spectral decomposition is shown in Figure 2, for HP, *AF_S_*, *VF_S_*, and *CSFF_S_*. The example shows that the LF and the HF components were clearly identified and quantified by the analysis. In this study, the VLF components appeared smaller than observed in many BBV studies [1,3,4,5,6], even for the HP BBV which is a standard series. This is due to the necessarily shorter period of 60 s observed compared to the usual 5 min. For this reason, we did not attempt to discuss the VLF power while also a detrend was introduced, further reducing the VLF components.

Group mean and standard deviation of *AF_S_*, *AF_D_*, *AF_M_*, *VF_S_*, *VF_D_*, *VF_M_*, *CSFF_S_*, *CSFF_D_* and *CSFF_M_* are presented in Table 1. 

*AF_M_* and *CSFF_M_* showed positive values (7.4 ± 1.7 mL/s and 0.1 ± 0.1 mL/s respectively), while *VF_M_* was negative (−7.2 ± 2.3 mL/s).

*VF_D_ P_TOT_* (0.662 ± 0.697 (mL/s)^2^) was significantly higher than *VF_S_* (0.322 ± 0.446 (mL/s)^2^, *p* < 0.001, ω = 0.837, dz = 0.556), and *AF_D_ P_TOT_* (0.230 ± 0.191 (mL/s)^2^, *p* = 0.004, ω = 0.956, dz = 0.692), and *CSFF_D_* (0.054 ± 0.048 (mL/s)^2^, *p* < 0.001, ω = 0.997, dz = 0.902). A sharp *P_TOT_* drop (3 to 10 times less) is noticed in all CSFF series, compared to AF (*p* = 0.035, ω = 0.862, dz = 0.531 for the systolic, and *p* < 0.001, ω = 0.999, dz = 1.023 for the diastolic peak comparison), and VF (*p* = 0.027, ω = 0.845, dz = 0.562, for the systolic, and *p* < 0.001, ω = 0.997, dz = 0.902 for the diastolic peak comparison). Higher *P_TOT_* was observed for *CSFF_S_* when compared to *CSFF_D_* (*p* = 0.012, ω = 0.784, dz = 0.518).

The diastolic *P_LF_* was significantly higher compared to the systolic one for both VF and AF, i.e., 31.8 ± 26.6 n.u. vs. 18.9 ± 25.3 n.u., *p* = 0.05, ω = 0.748, dz = 0.497; 45.2 ± 31.0 n.u. vs. 29.5 ± 27.9 n.u., *p* = 0.039, ω = 0.802, dz = 0.531 respectively. 

*P_HF_* for *CSFF_D_* was significantly higher than *P_HF_* for *AF_D_* (72.0 ± 21.1 n.u. vs. 45.3 ± 33.7n.u. *p* < 0.001, ω = 0.998, dz = 0.905) and *VF_D_* (72.0 ± 21.1 n.u. vs. 56.6 ± 27.7 n.u., *p* = 0.037, ω = 0.902, dz = 0.614).

The dz demonstrated a large or intermediate effect of the significant differences, with the exception of the comparison diastolic VF *P_LF_* vs. systolic VF *P_LF_*. The ω was always higher than 0.8 with the exception of diastolic VF *P_LF_* vs. systolic VF *P_LF_* and *CSFF_S_ P_TOT_* vs. *CSFF_D_ P_TOT_*, that had a power greater than 0.7.

The aimed unstressed resting condition was confirmed comparing mean HP and BF (obtained with the pulsoxymeter, the respiratory band, and the BBV analysis) between the two acquisitions (i.e., for blood and CSF). The mean HP (0.8 ± 0.1 s and 0.8 ± 0.2 s for blood and CSF acquisition respectively, HR = 75 beats/min), mean BF (0.27 ± 0.3 Hz for both the acquisitions), *f_HF_* (0.25 ± 0.06 Hz, 0.26 ± 0.05 Hz for the blood and CSF acquisitions, *p* = 0.391) did not significantly change. Additionally, a BBV power and LF/HF ratio (about 1/2) were found consistent with a mid-age population at rest.

### 3.3. Pulse Waves and the Influence of Breathing Phase

An example of the influence of the breathing phase on AF pulse wave analysis is shown in Figure 3. Complex distribution differences are observed between inspiration and expiration beats, with no clustering. Median shapes show a well-delineated pattern both when considering all cycles together independently of the breathing phase, and when segmenting inspiration and expiration beats.

An example of inspiration and expiration median pulse waves is displayed in Figure 4. Group means AF, VF, and CSFF amplitude computed when segmenting inspiration and expiration beats are reported in Table 2. 

No significant differences were observed between parameters computed for inspiration and expiration beats separately. However, a trend for higher absolute amplitude was found for systolic AF, mean AF, and systolic CSFF during inspiration.

## 4. Discussion

### 4.1. BBV Analysis

In this study, BBV was investigated from data derived with a novel non-invasive RT-PC-MRI sequence, which proved to be well tolerated by the participants and provided reliable BBV spectral analysis. Furthermore, the mean HP (derived from RT-PC-MRI data) and BF (measured with the thoracic belt) confirmed that scans were performed in unstressed resting condition, as an average HP = 0.8 s (equivalent to HR = 75 beats/min) and an average BF of about 0.25 Hz (equivalent to 15 breaths/min) were measured.

It is noteworthy that mean AF and VF values derived from RT-PC-MRI data were almost equal (i.e., within measurement errors), while the mean CSFF was close to zero, with a negligible group dispersion. This result, though expected in accordance to mass balance, provides evidence about the absence of significant biases in the quantification of AF, VF, and CSFF. However, possible scale biases between blood and CSF flows cannot be excluded due to the separate scans with different velocity encodings.

It is also worth mentioning that no flow inversions were observed for AF and VF, because data were acquired at a high cervical level (C1). The CSFF, conversely, oscillated around zero, given that our observation window was too short to observe the very slow CSFF downward from the IC to the spinal compartment. Indeed, such a continuous renewal of the CSFF, which is of utmost importance (e.g., in IC pressure regulation) was out of our scope [19]. 

One of the main results of this study is that AF, VF, and CSFF signals derived from RT-PC-MRI data clearly confirmed that the systolic pulse exerts a “water-hammer” effect in the IC compartment, which is a stiff volume requiring compensation by the VF and CSFF, as predicted by the well-established Monro–Kellie theory [24,25,26]. A rough insight of the volume compensation roles is provided by the comparison of systolic and diastolic mean pulse amplitudes of VF and CSFF (*VF_S_* − *VF_D_*: |−9.1 − (−5.9)| = 3.2 mL/s; *CSFF_S_* − *CSFF_D_*: |−1.8 − 1.4| = 3.2 mL/s), which on average are equal, suggesting a fifty-fifty distribution. A qualitative portrait of AF, VF, and CSFF in the systolic and the diastolic phases is provided in Figure 5 (left and right panel, respectively).

Passing to BBV waves, no substantial difference was found between *VF_D_* HF n.u. and *VF_S_* HF n.u, while *VF_D_* LF n.u. was significantly higher than *VF_S_* LF n.u., and *P_TOT_* was almost doubled when comparing *VF_D_* to *VF_S_*. This implies that the absolute power of HF *VF_D_* is about 2-fold higher in diastole than in systole, which can be readily explained by the thoracic pump [9,11], which increases the venous return during inspiration and it is likely to be mainly effective over the VF during diastole, when the flow rate is decreased. Remarkably, the HF respiratory waves were significantly predominant in the CSFF series both in systole and diastole, compared to LF. This highlights the buffer role of the CSF compartment, with large capacitive coupling with the venous compartment.

Conversely, the LF *VF_D_* power was 2-fold higher than in systole, a result that was not expected. A possible explanation might be in the diastolic increment of LF in the arterial compartment (*AF_D_)*. This consideration is highly suggestive of an LF modulation in the IC peripheral resistances, as highlighted in Figure 5. However, alternative contributions cannot be excluded, starting by a trivial influence of LF waves found in the systemic arterial pressure. Moreover, this result should be confirmed increasing the sample size: the observed power was 0.748 for the venous side, and the Bonferroni-corrected *p*-value was at the limit of the statistical significance (0.05).

Cues supporting the hypothesis of at least a partial contribution of brain vasomotion (i.e., modulation of peripheral resistances) at LF are: (i) systolic vs. diastolic changes almost parallel in AF and VF (see arrows marked “LF”, in Figure 5); (ii) the apparently independent CCSF BBV, fixed to HF prevalence. 

Indeed, if the observed AF BBV was fully driven by the systemic arterial pressure BBV (Figure 5, green arrow at the arterial input) and next transferred via the IC capacitive coupling to the VF and the CSF (similar to the main pulse waves) similar BBV PSD patterns and LF/HF ratios should be seen in the three flow signals. In addition, a transfer of LF waves via the IC arterio–venous capacitive coupling [14] would be rather more sensitive to the systolic AF modulation, since the major role of the capacitive coupling is compensating the systolic arterial water-hammer effect. Further evidence supporting this hypothesis is the poor presence of LF waves in the CSFF series [27]. Clearly, the CSF compartment is only capacitively coupled to AF and VF. More precisely, CSFF is capacitively driven by the instantaneous imbalance of (AF(t) + VF(t)) [15]. Conversely, the hypothesized parallel resistive coupling between AF and VF may bypass the CSF compartment. Further experiments aiming to test this hypothesis are warranted. Notably, greater knowledge of LF AF oscillation may impact treatment of neurodegenerative diseases. For instance, targeting naturally occurring vasomotion in patients with Alzheimer’s disease was suggested as a possible promising therapeutic option for preventing β-amyloid accumulation in the brain [28].

### 4.2. Pulse Wave Analyses and Respiratory Modulation

The normalized single-beat AF waves derived in this study showed very well-defined patterns compared to the rather smeared ones generally obtained with gated PC-MRI [8,15]. However, the most remarkable novel aspect introduced by the RT-PC-MRI is the possibility to observe the beat-by-beat dispersion of pulse waves, which gains further relevance when the inspiration and expiration phases are segmented. Additionally, the median wave that is obtained from RT-PC-MRI data is optimally delineated, as well as the inspiration/expiration ones.

The results presented in this study are limited to free breathing, which exerted a mild respiratory influence on AF, VF, and CSFF signals. Although no significant differences between inspiration and expiration were found in this study during free breathing, significant differences might be observed with different respiration patterns. Therefore, the ability to obtain clear median AF shapes of both inspiratory and expiratory beats with RT-PC-MRI could pave the way to a detailed analysis of the cerebrovascular effects of varied respiration patterns due to exercise and rehabilitation [29,30,31].

Despite the absence of significant differences, a trend for greater flows was found during inspiration. Higher mean VF during inspiration than expiration pinpointed the intrathoracic pressure as a major drive of venous return. Breathing phase affected CSFF pulse wave less than VF (i.e., delta with lower order of magnitude) but in the same direction as that of the VF. 

The observed higher AF during inspiration compared to during expiration could be due to Starling’s dependence of cardiac-output on venous-return or to the facilitation of arterial flow in phases of a better venous depletion.

### 4.3. Limitations

One of the main limitation of this study is that the evaluation of AF and VF was restricted to the ICAs and IJVs (left and right). However, this choice was due to the great difficulty in segmenting small collateral vein vessels with slow flow, mainly the spinal ones, and for avoiding signal artefacts. Indeed, the dynamic propagation of ROIs segmenting small vessels would not be sufficiently robust against subjects’ movements: since ROIs are propagated to all the time frames, even low movements would create a mismatch between an ROI and a small vessel. Nonetheless, ICAs are known to carry the majority and an approximately constant percentage of the total IC arterial flow [32], thus ICA AF is a good approximation of whole brain AP. Similar considerations can be made for the IJVs, which are the major veins for brain venous drainage in supine position. Different veins would have to be considered for brain venous drainage in the upright posture, as large drops in the IJV_s_% flow would occur [10,33]. However, the works investigating flows in the upright position had to use ultrasound [11] or low-field MRI scanners [34]. Second, we had to remove some signal parts because they were affected by artifacts. As an example, we had to remove the signal window shown in Figure 1 for both AF and VF: their flow rates had a sudden decrement, until around zero, because the subject moved during the RT-PC acquisition of the blood, so the vessels went outside the ROIs. However, as reported in the results (Section 3.1), a low percentage of the subjects (<15%) had this kind of artifact, and the artifact lasted less than 6 s over the 60 s-sequence, leaving enough signal for our analyses.

Third, given that two VENC were needed to acquire AF/VF and CSFF signals, two separate sequences had to be used for blood and CSF flows recording. Favoring subjects’ stationary conditions and limiting the current data analyses to amplitudes (power) of LF and HF modulations, as well as to breathing phase effects, permitted this limitation to be overcome. However, a deeper insight into the genesis of the described waves would require the check of time delays, phase relationships, and causal effects via multi-channel analyses [35]. At present acquiring a single sequence with a VENC suited to a wide spectrum of addressed flow velocities might be attempted, albeit with high technical difficulties.

Moreover, we could not perform a priori power analysis for dimensioning our sample size, because there is not a similar study in the literature, investigating arterial, venous, and CSF flow BBV as those we obtained from our RT-PC data. However, we decided to acquire 30 healthy volunteers starting from the observation that the published works using RT-PC for estimating arterial, venous, and CSF flow rates are usually limited to less than 20 healthy subjects [8,36,37,38,39,40]. Nevertheless, with our sample size, the effect of size demonstrated a large or intermediate effect of the significant differences, with the exception of the comparison of diastolic VF PLF vs. systolic VF PLF. The observed power was always higher than 0.8 with the exception of diastolic VF PLF vs. systolic VF PLF and *CSFF_S_*
*P_TOT_* vs. *CSFF_D_*
*P_TOT_*, which had a power greater than 0.7. With the averages and standard deviations of all the measures reported in Table 1 and Table 2, future works can dimension the group with a priori power analysis. 

The main topic addressed by the present study was BBV within flows at the highest possible cervical level (C1), i.e., the interface between the systemic CV and the IC vascular compartments. Free breathing is known to be the best condition permitting the separation of LF and HF components, conversely metronome or even forced breathing are known to have great influence on respiratory BBV patterns and amplitudes [41] as well as on LF-HF frequency [42]. Respiratory phase effects were addressed in this study during free-breathing, to focus on the performances of RT-PC-MRI in extracting single pulse cycles and in evaluating their dispersion. However, special breathing patterns will be explored in future analyses. 

When assessing autonomic modulations in the cerebrovascular compartment with MRI, a major limitation is head-up tilt (i.e., the most well-established non-invasive sympathetic stimulus, since decades), which is not applicable in commonly used MRI scanners. Therefore, other non-invasive stimuli (e.g., low body negative pressure) could be conceived to perform autonomic modulation during MRI scans. For instance, alternative non-invasive provocations could be chemoreceptive stimuli, such as hypercapnia, whose MRI-compatibility has already been tested and proved in previous studies including other kind of MRI sequences [43,44,45].

## 5. Conclusions

To the best of our knowledge, this study presented for the first time PC-RT-MRI applied to the assessment of intracranial AF, VF, and CSFF beat-by-beat variability. Although the investigation was limited to free-breathing conditions, the presence of both LF Mayer waves and HF respiratory waves was shown in the spectra derived from MRI data. The results presented in this study quantitatively confirm the great respiratory modulation, which is mainly relevant to the venous drainage and the arterial input. The latter is hypothesized to be influenced by the former, at least partially. Further studies to address causal relationships and the influence of breathing patterns are warranted.

## Figures and Tables

**Figure 1 biosensors-12-00417-f001:**
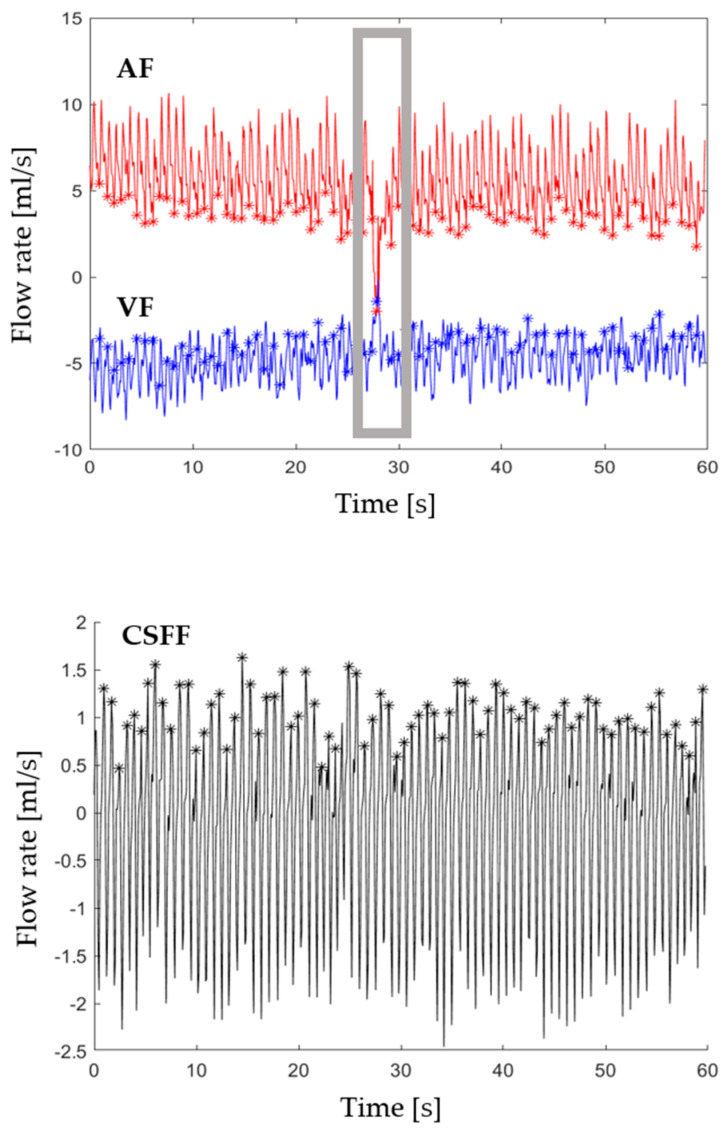
Example of AF (in red), VF (in blue), CSFF (in black) signals obtained after RT-PC-MRI data processing. AF and VF were derived from the same data, while CSFF was obtained from a separate sequence run. Diastolic fiducial points are marked by “*”. Diastolic fiducial points represented on AF signal correspond to AF minima. Diastolic fiducial points represented on VF signal are sampled according to AF minima, which are close but not coincident to VF maxima. Diastolic fiducial points represented on CSFF signal correspond to CSFF maxima. The short grey window in the AF and VF plots highlights a window of bad signal, which had to be discarded.

**Figure 2 biosensors-12-00417-f002:**
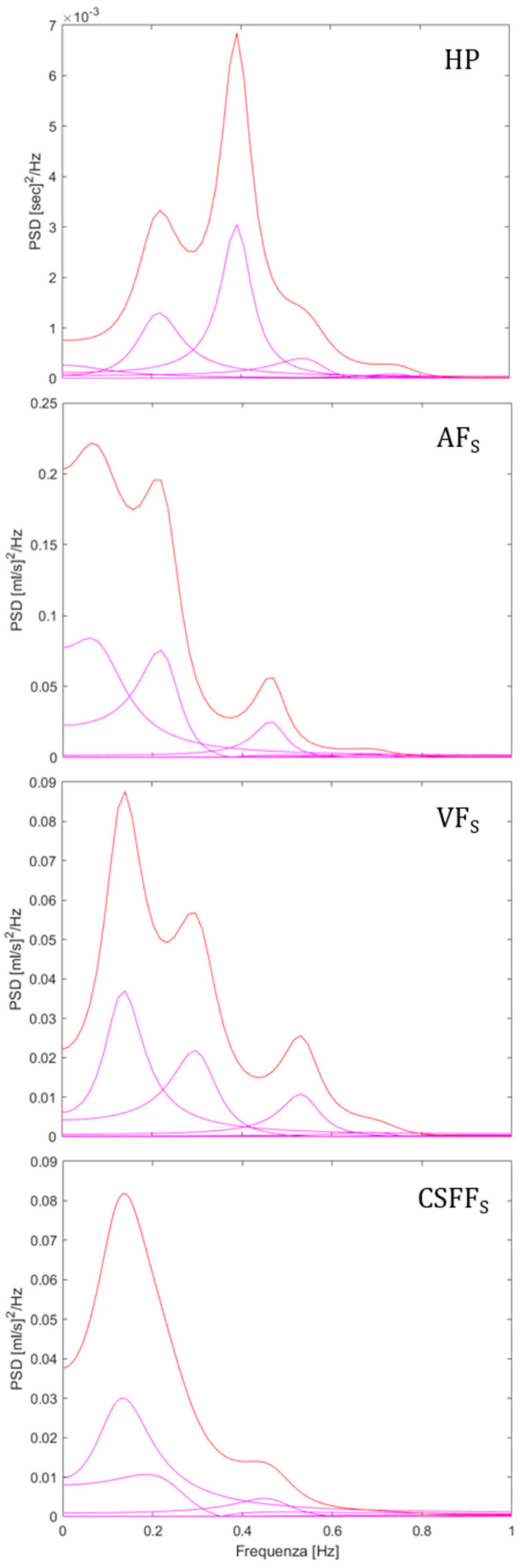
Example of AR PSD decomposition. From top to bottom: HP, *AF_S_*, *VF_S_*, and *CSFF_S_*. The represented spectra were computed from the MRI data of the same subject represented in Figure 1.

**Figure 3 biosensors-12-00417-f003:**
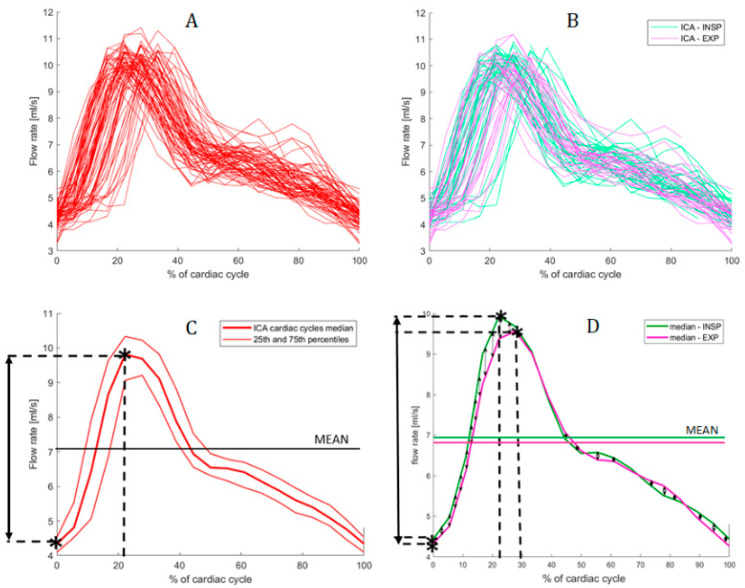
Example of median beat analysis on AF. (**A**) Normalized single-beat waves, from diastole to diastole. (**B**) Normalized single-beat waves during inspiration (represented in light cyan) and normalized single-beat waves during the expiration phase (represented in magenta). (**C**) Median beat and quartiles computed considering all single-beat waves, independently of breathing phase. (**D**) Median curve computed for inspiration (INSP, cyan) and expiration (EXP, magenta) median beats separately. Peak-to-peak amplitudes are highlighted in both (**C**,**D**) panels. The represented plots were computed from the MRI data of the same subject represented in Figure 1.

**Figure 4 biosensors-12-00417-f004:**
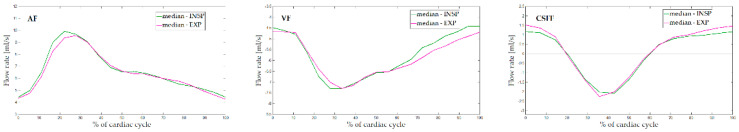
Example of inspiration (INSP, cyan) and expiration (EXP, magenta) median beats in AF, VF, and CSFF in one subject. The represented plots were computed from the MRI data of the same subject represented in Figure 1.

**Figure 5 biosensors-12-00417-f005:**
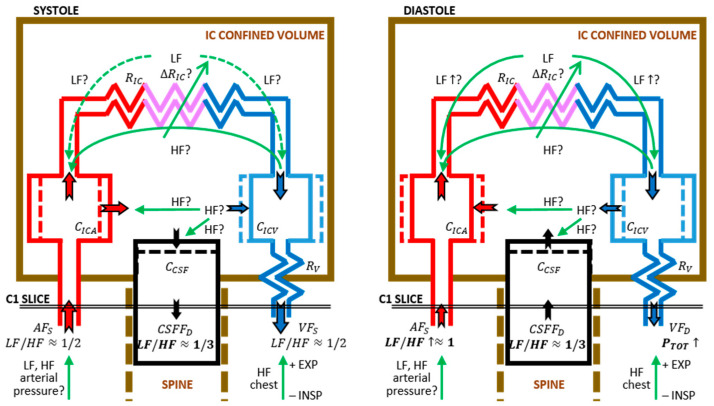
Schematic representation of systolic (**left panel**) and diastolic (**right panel**) flows, measured at C1 cervical level. AF is shown in red, VF is shown in blue, and CCSF is shown in black. The arrow dimension provides a qualitative sketch of flow rate order of magnitude. Flows are represented as hypothetically split into resistive and capacitive coupling flows. As to BBV, LF, and HF components, the main findings are reported in bold. Hypotheses about sources and propagation of LF and HF waves are indicated by the green arrows and labeled by question marks. Legend of symbols not in the text: *R_IC_* and *∆R_IC_*, peripheral IC resistances and their modulations; *C_ICA_* and *C_ICV_*, IC arterial and venous compliances, respectively; *C_CSF_*, IC CSF compliance, coupled to the spinal CSF compartment.

**Table 1 biosensors-12-00417-t001:** Results of AR PSD decomposition. Values are reported as average ± std in the group of 30 healthy subjects. Significant post-hoc contrasts (*p* < 0.05, two-way repeated measure-ANOVA) are noted with couples of letters (a–k). The *p*-value, effect size estimate Cohen’s dz and the observed power (ω) of each significant post-hoc comparison are reported in the legend.

Series	Mean	$P_{TOT}$	$f_{LF}$	$P_{LF}$	$f_{HF}$	$P_{HF}$
	[s]	[s]^2^	[Hz]	[n.u.]	[Hz]	[n.u.]
HP	0.8 ± 0.1	0.0089 ± 0.0124	0.12 ± 0.01	21.0 ± 24.0	0.25 ± 0.06	61.8 ± 20.1
$HP_{\mathrm{CSF}}$	0.8 ± 0.2	0.0077 ± 0.0087	0.10 ± 0.03	16.1 ± 22.6	0.26 ± 0.05	57.1 ± 18.4
	[mL/s]	[mL/s]^2^	[Hz]	[n.u.]	[Hz]	[n.u.]
$AF_{S}$	11.1 ± 2.6	0.389 ± 0.598 ^d^	0.10 ± 0.02	29.5 ± 27.9^i^	0.26 ± 0.06	54.3 ± 26.2
$AF_{D}$	5.1 ± 1.4	0.230 ± 0.191 ^b,e^	0.10 ± 0.02	45.2 ± 31.0 ^i^	0.26 ± 0.07	45.3 ± 33.7 ^k^
$VF_{S}$	−9.7 ± 3.4	0.322 ± 0.446 ^a,f^	0.09 ± 0.03	18.9 ± 25.3 ^j^	0.28 ± 0.05	58.0 ± 24.0
$VF_{D}$	−6.0 ± 2.1	0.662 ± 0.697 ^a,b,c,g^	0.09 ± 0.03	31.8 ± 26.6 ^j^	0.26 ± 0.05	56.6 ± 27.7 ^l^
$CSFF_{S}$	−2.3 ± 0.7	0.091 ± 0.082 ^d,f,h^	0.10 ± 0.02	23.7 ± 24.8	0.26 ± 0.05	64.4 ± 25.4
$CSFF_{D}$	1.4 ± 0.5	0.054 ± 0.048 ^c,e,g,h^	0.10 ± 0.03	26.8 ± 26.0	0.25 ± 0.05	72.0 ± 21.1 ^k,l^

Legend: ^a^
*p*-value < 0.001, dz = 0.556, ω = 0.837; ^b^
*p* = 0.004, ω = 0.956, dz = 0.692; ^c^
*p* < 0.001, ω = 0.997, dz = 0.902; ^d^
*p* = 0.035, ω = 0.862, dz = 0.531; ^e^
*p* < 0.001, ω = 0.999, dz = 1.023; ^f^
*p* = 0.027, ω = 0.845, dz = 0.562; ^g^
*p* < 0.001, ω = 0.997, dz = 0.902; ^h^
*p* = 0.012, ω = 0.784, dz = 0.518; ^i^
*p* = 0.039, ω = 0.802, dz = 0.531; ^j^
*p* = 0.05, ω = 0.748, dz = 0.497; ^k^
*p* < 0.001, ω = 0.998, dz = 0.905; ^l^
*p* = 0.037, ω = 0.902, dz = 0.614.

**Table 2 biosensors-12-00417-t002:** Mean amplitudes in median pulse waves [mL/s]. Legend: AF = arterial flow; VF = venous flow; CSFF = cerebrospinal fluid flow; Syst = systolic; Dia = diastolic; Insp = inspiration; Exp = expiration; Insp = inspiration.

Signal	All	Insp	Exp	∆ Insp-Exp	*p*-Value of ∆
Mean AF	7.39 ± 1.72	7.32 ± 1.65	7.26 ± 1.64	0.0607	0.068
Mean VF	−7.28 ± 2.28	−7.35 ± 2.45	−7.25 ± 2.23	−0.0954	0.216
Mean CSFF	0.111 ± 0.134	0.085 ± 0.098	0.093 ± 0.134	−0.0084	0.772
Syst AF	10.29 ± 2.44	10.20 ± 2.42	10.08 ± 2.39	0.1244	0.059
Syst VF	−9.08 ± 3.14	−9.21 ± 3.35	−9.16 ± 3.17	−0.0493	0.524
Syst CSFF	−1.81 ± 0.60	−1.87 ± 0.60	−1.80 ± 0.62	−0.0733	0.062
Dia AF	5.11 ± 1.37	5.02 ± 1.26	5.02 ± 1.26	0.0063	0.875
Dia VF	−5.90 ± 2.05	−5.96 ± 2.13	−5.88 ± 2.08	−0.0830	0.310
Dia CSFF	1.44 ± 0.53	1.38 ± 0.42	1.41 ± 0.42	−0.0267	0.416

## Data Availability

The data presented in this study are available on request from the corresponding author.

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
