# Peer review of "Real-Time Phase-Contrast MRI to Monitor Cervical Blood and Cerebrospinal Fluid Flow Beat-by-Beat Variability"

_biosensors, 2022, doi:10.3390/bios12060417_

Round 1

Reviewer 1 Report

Generally, the topic and contents of manuscript are quite interesting and highly relevant to the scope of the Biosensors. It is quite novel that authors described the PC-RT-MRI approach for the first time to assess the BBV. The results and conclusion are quite clear. The limitations of the study have been also addressed.

But its acceptance, the manuscript needs to be minor revised.

 Comments:

  1. Line 37, delete variability after BBV.
  2. Describe the approach of Doppler echography and its limitations in the introduction section rather than in the discussion section, (move some paragraphs of 4.1 section to introduction section).
  3. line 187, perhaps authors can explain a bit in the discussion section about why bad signal interval occur and need to be discarded?  
  4. line 202, explain “VLF instead looks smaller”

Author Response

We thank the reviewers and the Academic Editor for their valuable comments. Hereafter the answers to reviewer’s comment (in red). In the manuscript, the revisions are in red and we added a comment for an immediate understanding of which comment we refer to (Reviewer 1 – R1; Reviewer 2 – R2; Academic Editor – AE; Question#- Q#).

Generally, the topic and contents of manuscript are quite interesting and highly relevant to the scope of the Biosensors. It is quite novel that authors described the PC-RT-MRI approach for the first time to assess the BBV. The results and conclusion are quite clear. The limitations of the study have been also addressed.

But its acceptance, the manuscript needs to be minor revised.

Comments:

R1Q1 - Line 37, delete variability after BBV.

Response:

Thank you, we’ve removed it

R1Q2 - Describe the approach of Doppler echography and its limitations in the introduction section rather than in the discussion section, (move some paragraphs of 4.1 section to introduction section).

Response:

Thank you for the suggestion. We moved and adapted the paragraphs from the discussion to the introduction section.

R1Q3 - line 187, perhaps authors can explain a bit in the discussion section about why bad signal interval occur and need to be discarded?

Response:

To make this point clearer, we made the following changes: first, we deepened this aspect in the methods; second, we changed Figure 1 of the results, choosing a signal with a clearer artifact; third, we expanded the discussion; and finally we added and commented the topic in the Limitations section.

R1Q4 - line 202, explain “VLF instead looks smaller”

Response:

Thank you for pointing this out. We have rephrased the sentence that was not clear.

Reviewer 2 Report

The manuscript presents an interesting experimentation.

The number of subjects enrolled for the definition of the model is low and this implies a low statistical significance of the data.

Is it possible to increase the number of subjects enrolled?

Author Response

We thank the reviewers and the Academic Editor for their valuable comments. Hereafter are the answers to the reviewer’s comment (in red). In the manuscript, the revisions are in red and we added a comment for an immediate understanding of which comment we refer to (Reviewer 1 – R1; Reviewer 2 – R2; Academic Editor – AE; Question#- Q#).

Reviewer 2

The manuscript presents an interesting experimentation.

The number of subjects enrolled for the definition of the model is low and this implies a low statistical significance of the data.

Is it possible to increase the number of subjects enrolled?

Response:

To answer this comment and as suggested by the Academic Editor, we have performed a power analysis.

On the premise, to the best of our knowledge, the published works using RT-PC for estimating arterial, venous and CSF flow rates are usually limited to less than 20 healthy subjects [1-6]. However, there is not a similar study in literature, investigating arterial, venous and CSF flow beat-to-beat variability as those we obtained from the RT-PC data. Since preliminary data such as those reported by our study are missing in the literature, we could not perform an a priori power analysis for dimensioning our sample group of healthy volunteers. However, after reading the reviewer’s and the Academic Editor’s comments, we acknowledge that it is important to show and discuss the significance of our statistics with an observed power analysis. For this reason, we have added the observed power analysis and the effect size of the significant tests. With our sample size (30 subjects), the effect size demonstrated a large or intermediate effect of the significant differences, except for the comparison of diastolic VF PLF vs systolic VF PLF. The observed power was always higher than 0.8 except for diastolic VF PLF vs systolic VF PLF and CSFFS PTOT vs CSFFD PTOT, however having a power greater than 0.7. We reported the averages and standard deviations of all the measures in Table 1 and Table 2, so future works can use these data for dimensioning the group with a priori power analysis.

We have revised the methods, results, and discussion accordingly.

  1. Dreha-Kulaczewski, S.; Joseph, A.A.; Merboldt, K.D.; Ludwig, H.C.; Gartner, J.; Frahm, J. Inspiration is the major regulator of human CSF flow. J Neurosci 2015, 35, 2485-2491, doi:10.1523/JNEUROSCI.3246-14.2015.
  2. Yildiz, S.; Thyagaraj, S.; Jin, N.; Zhong, X.; Heidari Pahlavian, S.; Martin, B.A.; Loth, F.; Oshinski, J.; Sabra, K.G.J.J.o.M.R.I. Quantifying the influence of respiration and cardiac pulsations on cerebrospinal fluid dynamics using real‐time phase‐contrast MRI. 2017, 46, 431-439.
  3. Ohno, N.; Miyati, T.; Noda, T.; Alperin, N.; Hamaguchi, T.; Ohno, M.; Matsushita, T.; Mase, M.; Gabata, T.; Kobayashi, S.J.D. Fast Phase-Contrast Cine MRI for Assessing Intracranial Hemodynamics and Cerebrospinal Fluid Dynamics. 2020, 10, 241.
  4. Aktas, G.; Kollmeier, J.M.; Joseph, A.A.; Merboldt, K.-D.; Ludwig, H.-C.; Gärtner, J.; Frahm, J.; Dreha-Kulaczewski, S.J.F.; CNS, B.o.t. Spinal CSF flow in response to forced thoracic and abdominal respiration. 2019, 16, 1-8.
  5. Kollmeier, J.M.; Gürbüz-Reiss, L.; Sahoo, P.; Badura, S.; Ellebracht, B.; Keck, M.; Gärtner, J.; Ludwig, H.-C.; Frahm, J.; Dreha-Kulaczewski, S.J.S.r. Deep breathing couples CSF and venous flow dynamics. 2022, 12, 1-13.
  6. Dreha-Kulaczewski, S.; Joseph, A.A.; Merboldt, K.D.; Ludwig, H.C.; Gartner, J.; Frahm, J. Identification of the Upward Movement of Human CSF In Vivo and its Relation to the Brain Venous System. J Neurosci 2017, 37, 2395-2402, doi:10.1523/JNEUROSCI.2754-16.2017.

Round 2

Reviewer 2 Report

Nothing